# Research and Optimization of High-Performance Front-End Circuit Noise for Inertial Sensors

**DOI:** 10.3390/s24030805

**Published:** 2024-01-26

**Authors:** Yuzhu Chen, Xin Liu, Longqi Wang, Tao Yu, Zhi Wang, Ke Xue, Yanlin Sui, Yongkun Chen

**Affiliations:** 1Changchun Institute of Optics, Fine Mechanics and Physics, Chinese Academy of Sciences, Changchun 130033, China; chenyz_mail@163.com (Y.C.); yut@ciomp.ac.cn (T.Y.); suiyanlin@ciomp.ac.cn (Y.S.);; 2School of Fundamental Physics and Mathematical Sciences, Hangzhou Institute for Advanced Study, UCAS, Hangzhou 310024, China

**Keywords:** transformer, capacitive displacement sensor, JFET, capacitive sensing resolution

## Abstract

An inertial sensor is a crucial payload in China’s Taiji program for space gravitational wave detection. The performance of the capacitive displacement sensing circuit in the low-frequency band (0.1 mHz to 1 Hz) is extremely important because it directly determines the sensitivity of the space gravitational wave detection missions. Therefore, significant, yet challenging, tasks include decreasing the low-frequency noise in capacitive displacement sensing circuits and improving the capacitive sensing resolution. This study analyzes the noise characteristics of the pre-amplifier circuit within the capacitive sensing circuit, achieves precise tuning of the transformer bridge, and examines how transformer parameters affect noise. In addition, this study introduces a method using a discrete JFET to reduce the operational amplifier current noise and analyzes how feedback resistance and capacitance in TIA circuits affect the overall circuit noise. The proportional relationship between different transformer noises and TIA noise before and after optimization was analyzed and experimentally verified. Finally, an optimized TIA circuit and a superior transformer were utilized to achieve an increase in the capacitive sensing resolution from 1.095 aF/rtHz @ 10 mHz to 0.84 aF/rtHz @ 10 mHz, while improving the performance by 23%. These findings provide valuable insights into further decreasing circuit noise and increasing the capacitive sensing resolution.

## 1. Introduction

With the first detection of gravitational waves in 2015 by the ground-based gravitational wave observatory LIGO [1], gravitational wave detection has gradually become a popular topic and an academic frontier. It is expected that gravitational wave detection and research will provide an irreplaceable method to reveal the nature of gravity and space-time, accurately test Einstein’s general relativity theory, detect new physics beyond its theory, explore the unified field theory and the origin of the universe, and related new phenomena [2]. China’s Taiji Program for Space Gravitational Wave Detection achieved a significant milestone on August 31, 2019, with the successful launch of the experimental satellite Taiji No. 1 and subsequent in-orbit testing that yielded remarkable results [3].

In the Taiji project, an inertial sensor served as the core component and provided an inertial reference for gravitational wave detection. To achieve high accuracy in the residual acceleration noise measurements, it is crucial to optimize the performance of the capacitive displacement sensing circuit, which forms the backbone of the entire system [4,5,6].

Capacitive displacement sensors are widely used for measuring relative displacement. The method has been extensively applied in scientific experiments such as the verification of key technologies for space gravitational wave detection [7], equivalent principle validation, the measurement of earth’s gravity field [8], non-drag flight confirmation [9], obtaining information on the relative position between the test mass and a spacecraft, and acceleration under external non-conservative forces [10,11,12,13,14].

LISA Pathfinder, the European Space Agency’s technology demonstrator mission for future spaceborne gravitational wave observatories, was launched on 3 December 2015, from the European space port of Kourou, French Guiana [15]. It was the highest precision satellite experiment mission ever tested in space flight, with final test results of 0.7–1.8 af from in-flight experiments [16]. The noise performance of LISA Pathfinder has further improved because of reduced Brownian noise and the availability of numerous long noise measurement runs, during which no perturbation is purposely applied to the test masses. The noise levels and frequencies are now well beyond the original goals of LISA Pathfinder, and even below the LISA [17]. Davor Mance describes the design process of a preliminary engineering prototype of LISA Pathfinder, which provides an excellent capacitive sensing scheme [18]. Reference [19] describes the process and results of our previous development of capacitive sensing circuits, which reached 1.095 aF/rtHz at 10 mHz. In this paper, the transformer is optimized on the basis of our previous research, and a discrete TIA circuit is used to reduce the circuit noise.

The inertial sensor system first converts the tiny displacement into a tiny capacitance tolerance through the sensitive structure and then realizes the measurement of the tiny tolerance via the capacitive sensing circuit. In a manner that differs from other capacitive sensing circuits, capacitive sensing circuits used for space gravitational wave detection with inertial sensors require a high capacitive sensing resolution in the low frequency band of 0.1 mHz–1 Hz. Therefore, noise analysis and circuit optimization of the capacitive sensing circuit in the low frequency band of 0.1 mHz–1 Hz are the key to improving the resolution of capacitive sensing.

The tiny capacitance tolerance causes an imbalance of the currents inside the transformer’s primary winding, which generates a differential voltage on the transformer’s secondary winding. This voltage is amplified by trans-impedance amplifier (TIA) and AC amplifier circuit, band-pass filtered, and then a demodulation signal consistent with the excitation signal frequency (100 kHz) is employed to convert the AC carrier signal containing tolerance information into a DC differential voltage. Only the frequency component in phase with the injection frequency is extracted using synchronous demodulation [18]. The analog signal is finally converted to a digital form by an analog-to-digital converter. This facilitates the high-precision assessment of residual acceleration noise in the inertial sensors for space gravitational wave detection. A schematic of the capacitive sensing circuit is shown in Figure 1.

The resolution of the capacitive sensing circuit is primarily influenced by the performance of its front-end circuit, because any noise present in this stage is amplified and transmitted along the link to the data acquisition end. The equivalent noise of the transformer was simulated and analyzed in this study. Additionally, a discrete TIA circuit structure utilizing a junction field-effect transistor (JFET) as the input stage was adopted to mitigate the noise based on the noise model of the TIA circuit. Furthermore, the influential factors affecting the noise performance of the discrete TIA circuit were analyzed. The noise generated by various transformers and discrete TIA circuits was empirically measured, and the experimental results were found to be consistent with theoretical calculations. Experimental validation confirmed the capacitive sensing resolution employed for low-frequency space gravitational wave detection. Compared with the previous test results using a planar transformer and non-discrete TIA circuit, the resolution of the capacitive sensing was further optimized from 1.095 aF/rtHz [19] to 0.84 aF/rtHz. This study offers valuable insights for further mitigating the circuit noise and enhancing the resolution of capacitive sensing, thereby contributing to advancements in reducing technical interference and optimizing measurement precision.

## 2. Transformer Bridge Noise Analysis

The transformer bridge circuit is a crucial component in capacitive displacement sensors, which are utilized in microdisplacement detection circuits to convert small changes in displacement into capacitance variations and to enable precise measurements.

### 2.1. Working Principle of Transformer Bridge

The front-end amplifier circuit of the capacitive sensing system primarily comprises of a transformer bridge and a TIA circuit, as shown in Figure 2.

The test mass (TM) and two-electrode plates of the sensor constitute capacitor $C_{1}$ and capacitor $C_{2}$, which is referred to as a nominal capacitor when the TM is in the central position, such that $C_{1}=C_{2}=C_{0}$, and $C_{0}$ denotes a nominal capacitor. When the TM moves in parallel between the two parallel electrode plates, a capacitance difference between the two plates is generated such that the sensing bridge and the current in the two primary windings of the transformer are unbalanced. The presence of an unbalanced current in the primary windings of the transformer induces a differential voltage in the secondary windings, which is directly proportional to the gain of the sensing circuit and the displacement of the TM relative to the center position between the two plates. The TIA circuit will further amplify the signal. Specifically, $C_{t1}$ and $C_{t2}$ are tuning capacitors, and the circuit operates at the resonant frequency to minimize noise by adjusting the size of the tuning capacitor [14]. In addition, $C_{a1}$ and $C_{a2}$ are drive capacitors, $C_{q}=2(C_{t}+C_{0})$. The transformer incurs a loss tan⁡δ, $\tan\delta =\frac{R_{L}}{\omega L}\approx \delta =\frac{1}{Q}$, where Q denotes the transformer’s quality factor and L denotes the transformer’s inductance. The expression for the actual inductance is given by $L_{R}=L\left(1-j\delta \right)$.

The expression for the actual impedance is given in Equation (1).
(1)$$Z_{BR}=\frac{R_{L}+j\omega L}{1+KR_{L}\frac{C_{a}C_{q}}{C_{a}+C_{t}}j\omega +KL\frac{C_{a}C_{q}}{C_{a}+C_{t}}{\left(j\omega \right)}^{2}}$$

Equation (2) expresses the thermal noise of the resonant bridge in terms of the equivalent impedance [9]
(2)$$e_{BR}=\frac{2\sqrt{4k_{B}T{ℜ}_{Z_{BR}}}}{\omega C_{FB}\left|Z_{BR}\right|}$$
where $k_{B}$ denotes the Boltzmann constant, T denotes absolute temperature, and ${ℜ}_{Z_{BR}}$ denotes the real part of $Z_{BR}$.
(3)$${ℜ}_{Z_{BR}}=\frac{R_{L}}{1-\omega^{2}KL\frac{C_{a}C_{q}}{C_{a}+C_{t}}\left(2-\omega^{2}KL\frac{C_{a}C_{q}}{C_{a}+C_{t}}\left(1+{\left(\frac{R_{L}}{\omega L}\right)}^{2}\right)\right)}$$

The output impedance is affected by the inductance and quality factors of the transformer.

### 2.2. Analysis of the Transformer Parameters

The transformer bridge plays a crucial role in the capacitive sensing circuit because its parameters directly affect the resonant frequency, equivalent output impedance, and output voltage noise of the bridge circuit. The specific parameters for both the transformer bridge and TIA are listed in Table 1.

The output voltage noise of the circuit is affected by the resonant frequency. Therefore, when comparing the effect of both transformers on the circuit, it is essential to initially adjust the tuning capacitance and align the circuit to a resonant frequency of 100 kHz in order to mitigate the output voltage noise [20].

When the quality factor exceeds 100, for a fixed resonant frequency, the primary determinant of the resonant frequency corresponds to the inductance of the transformer. In order to ensure that the resonant frequency is aligned with the target frequency and minimizes output voltage noise, adjustments to the resonant capacitance are necessary when dealing with transformers with varying inductance values. When the imaginary part of the bridge impedance equals zero, the solution corresponds to the resonant frequency and is calculated in Equation (4) as follows [18]:(4)$$1-2\omega^{2}C_{q}\frac{C_{a}}{C_{a}+C_{t}}(1+\delta^{2})=0$$

The simulated and analyzed output impedances of the bridge circuit corresponding to transformers 1 and 2 with different parameters are given. At a frequency of 100 kHz, the bridge’s equivalent impedance for the two transformers reaches significantly different maximum values, namely 544.3 kΩ and 1085 kΩ, respectively. The impedance exhibited a rapid decrease when the frequency deviates from 100 kHz. The output voltage noise of the bridge and the TIA were determined using Equation (2). Figure 3b shows the output voltage noise when transformers 1 and 2 were employed.

The output voltage noise of the transformer 1 circuit is measured at 179.8 nV/rtHz, while that for the transformer 2 circuit it is observed to be lower, at 129.2 nV/rtHz, thereby indicating a difference in their respective noise curves. Furthermore, the change in noise for transformer 1 exceeded that for transformer 2. Hence, it is important to carefully consider the effect of the transformer parameters on the overall circuit performance.

## 3. Analysis and Optimization of TIA Noise Influencing Factors

The TIA circuit serves as the active amplifier circuit in the front stage of the capacitive sensing circuit and plays a pivotal role in the overall sensing link by providing the highest signal gain. Its effect on the capacitive sensing resolution is extremely important and necessitates the analysis and optimization of TIA noise.

### 3.1. TIA Circuit Noise Analysis and Optimization

#### 3.1.1. TIA Circuit Noise Analysis

The TIA circuit noise model is shown in Figure 4, where the noise term encompasses the amplifier voltage noise, the amplifier current noise, and the thermal noise arising from the feedback resistance and capacitance. Given that the evaluation band for the performance index of the capacitive sensing circuit within the space gravitational wave detection system ranges from 0.1 mHz to 1 Hz, it is important to consider the system parameters pertaining to the low frequency data.

The noise gain of an op amp is equivalent to its non-inverting signal gain:(5)$$NG=2\times \frac{Z_{FB}}{{ℜ}_{BR}}+1$$
where ${ℜ}_{BR}$ denotes the real part at the resonant frequency point of $Z_{BR}$. The primary circuit parameters are listed in Table 2.

$ℜ\left[Z_{FB}\right]$ denotes the real part of $\left|Z_{FB}\right|$, and the calculations are given in Equations (6) and (7):(6)$$ℜ\left[Z_{FB}\right]=\frac{R_{FB}}{1+{\left(\omega R_{FB}C_{FB}\right)}^{2}}$$
(7)$$\left|Z_{FB}\right|=\frac{R_{FB}}{\sqrt{1+{\left(\omega R_{FB}C_{FB}\right)}^{2}}}$$

As shown in Table 2, the noise contribution of the TIA circuit originates primarily from the current noise of the operational amplifier. Decreases in the current noise can significantly decrease the overall TIA circuit noise. However, an operational amp chip has not been identified as capable of substantially reducing the TIA current noise given the performance limitations of existing integrated op amp processes and aerospace-grade chip restrictions in space gravitational wave detection projects. Therefore, [18] suggested using a discrete JFET as the input stage for the TIA circuit to short-circuit the operational amplifier current noise and decrease the overall TIA circuit noise.

#### 3.1.2. Discrete TIA Noise Analysis

The discrete TIA employs a JFET as the input stage of its circuit. The JFET exhibited an exceptionally high input impedance, and thus it is extremely suitable for implementation as an input stage.

Figure 5 shows the optimized circuit configuration. By introducing the JFET, the current noise of the operational amplifier was effectively reduced, thereby allowing only the current noise inherent to the JFET to propagate through the feedback resistance and capacitance.

The noise observed when utilizing the JFET as the input stage comprised of the amplifier voltage noise, the amplifier current noise, the TIA thermal noise, the JFET voltage noise, the JFET current noise, and the JFET thermal noise. The components are listed in Table 3. By utilizing the JFET as the input stage, the amplifier’s input current noise was substituted by the JFET’s input current noise, although a new noise term associated with the JFET was introduced. We consider U440 as an example, and its current noise index is merely 4.2 fA/rtHz, which is significantly lower than the equivalent current input noise of the op amp. However, $r_{DS}$ in the noise model described in [18] may be more accurately characterized by $R_{ds}$ [21]:(8)$$R_{ds}={\left(K_{d}g_{m}\right)}^{-1}$$

The coefficient $K_{d}$, which is associated with the form, size, and bias of the JFET, exhibits a value of approximately 1 in the linear region under normal operating conditions and approximately 0.67 in the saturated region [21]. In addition, $g_{m}$ denotes the transconductance of the JFET, and the calculated $R_{ds}$ was observed as 158 Ω, which is similar to the calculated result of 100 Ω in extant studies [18]. The contribution of the noise component to the overall noise was relatively low. Hence, it is observed that the two factors are comparable and do not warrant extensive discussion.

The data presented in Table 3 demonstrate that the utilization of the JFET as the input stage in the TIA circuit effectively mitigated the impact of the equivalent amplifier circuit noise, thereby significantly decreasing the TIA circuit noise [18].

The use of a JFET not only reduces the noise in the TIA circuit, but also reduces the performance requirements of the operational amplifiers in the circuit. This makes it easier to select devices that meet aerospace requirements in engineering. However, the differences in the performance of the JFET will cause the circuit performance and symmetry to be reduced. After screening and testing 60 JFETs, two JFETs with the most similar performance were selected for experimental testing. In addition, we have carried out domestic research work to avoid restrictions on purchase channels.

### 3.2. Discrete TIA Noise Influencing Factors

The TIA noise was affected by the JFET, op amp, feedback resistance, and capacitance. In this section, the effect of each component on the noise characteristics of the TIA circuit is analyzed. The effect of novel discrete JFET driving modes is explored in a future study.

#### 3.2.1. Effect of Feedback Capacitance

As listed in Table 3, the equation for TIA noise is given as follows:(9)$$N\_TIA=\sqrt{\frac{CR_{FB}^{2}+8k_{B}TR_{FB}}{1+{\left(\omega R_{FB}C_{FB}\right)}^{2}}+\frac{4BR_{FB}}{{ℜ}_{BR}\left(1+{\left(\omega R_{FB}C_{FB}\right)}^{2}\right)}+B}$$

The constants B and C are chip performance-related parameters.
(10)$$B=2u_{AMP}^{2}+4i_{AMP}^{2}r_{DS}^{2}+4u_{FET}^{2}+2i_{FET-}^{2}r_{DS}^{2}+16k_{B}Tr_{DS}$$
(11)$$C=\frac{4B}{{ℜ}_{BR}^{2}}+2\times i_{FET-}^{2}$$

Figure 6 shows the variation of TIA noise with respect to $C_{FB}$ described by Equation (9). As $C_{FB}$ is only present in the denominator, the TIA noise exhibits an inverse relationship with $C_{FB}$‘s magnitude.

Although increases in the feedback capacitance can effectively decrease the TIA noise, it can also lead to a decrease in gain. In multistage amplification circuits, priority is typically given to the amplification of the first-stage system. However, this conflicts with the output noise of the TIA and thus requires comprehensive consideration in practical applications.

#### 3.2.2. Effect of Feedback Resistance

By analyzing Equation (9) or examining Figure 6, it is evident that a correlation exists between the TIA noise and $C_{FB}$. Specifically, the TIA noise decreases when $C_{FB}$ increases. The variation in $R_{FB}$ also affects the noise generated in TIA circuits.

However, it is important to note that the TIA circuit must ensure proper amplification of the capacitor with the current flowing through the feedback capacitor as opposed to the feedback resistance. Hence, the impedance of the feedback capacitor must be significantly lower than that of the feedback resistor. These requirements are generally fulfilled by $Z_{C_{FB}}<10\times Z_{R_{FB}}$. The variation in the TIA noise is shown in Figure 7.

The effects of $Z_{FB}$ and NG can be disregarded. Thus, the TIA noise component is primarily affected by $ℜ\left[Z_{FB}\right]$, which aligns with the dominant contribution of the TIA thermal noise in the circuit, as listed in Table 3.

Therefore, TIA circuit noise can be mitigated by appropriately increasing feedback resistance.

#### 3.2.3. Effect of the Transformer Bridge

The discussion on TIA circuit noise in the previous sections was limited to the zero-input state. However, in practical applications, the contribution from the transformer noise is an indispensable component. The front end of the TIA circuit is connected to the transformer, and thermal noise generated by the transformer bridge significantly affects the noise characteristics of the TIA circuit. The input noise generated by the transformer is given in Equation (2).

Based on the simulation and analysis of transformer noise in Section 2.2, the optimal equivalent noise level for the transformer utilized by the current experimental team is determined as 129 nV/rtHz. As shown in Table 3, the TIA noise reaches a maximum of 34.71 nV/rtHz, which is less than one-third of the transformer noise. The contribution from the transformer noise significantly exceeds that of the overall TIA circuit noise. Furthermore, when considering the noise from the transformer bridge, Equation (12) is derived to represent TIA noise:(12)$$N\_TIA=\sqrt{\frac{CR_{FB}^{2}+8k_{B}TR_{FB}}{1+{\left(\omega R_{FB}C_{FB}\right)}^{2}}+\frac{4BR_{FB}+16k_{B}TR_{FB}^{2}}{{ℜ}_{BR}\left(1+{\left(\omega R_{FB}C_{FB}\right)}^{2}\right)}+B}$$

Given the performance limitations of transformers, the thermal noise generated by the equivalent resistance of the transformer bridge constitutes the primary source that contributes to the TIA noise in the overall TIA noise analysis. After considering the effect of the transformer noise, Figure 8 shows the correlation between the TIA noise and the feedback resistance and feedback capacitance.

The effect of adjusting the feedback resistance or feedback capacitance on the TIA noise was constrained. The decrease in the TIA noise is no longer significant given the effect of the transformer bridge noise on the change in feedback resistance. A noise variation of 6 nV was considered negligible in this study.

Decreases in the feedback capacitance in the TIA can effectively mitigate its noise. However, the adjustment also alters the gain of the TIA and does not significantly contribute to reducing the TIA noise after normalization. Decreases in the feedback resistance can potentially mitigate the TIA noise. However, given the substantial contribution of the equivalent input noise from the transformer, the effect of feedback resistance on the noise becomes negligible. The noise of the capacitive sensing front-end amplifier circuit is determined by the transformer bridge and the TIA. The TIA noise was effectively reduced by employing a discrete JFET as the input stage of the TIA, thereby making the input noise of the transformer the primary concern. Hence, the performance of the transformer becomes a critical limiting factor in determining system noise. Figure 9 summarizes the proportional relationship between transformer noise and TIA noise. It can also be seen that after the use of a discrete TIA, the equivalent noise generated by the transformer is the main body of noise, accounting for about 80% of the total noise.

## 4. Experimental Verification and Discussion

### 4.1. Noise Testing of Transformer Bridges

We augment the analysis of the transformer noise in Section 2.2, and the TIA circuit noise in Section 3.1, in a TIA circuit that does not utilize a JFET as the input stage. The noise at the TIA output for transformer 1 was measured as 292 nV/rtHz, while transformer 2 exhibited a slightly lower value of approximately 264 nV/rtHz, accounting for approximately 90% of the noise generated by transformer 1.

The noise test results of transformers 1 and 2 after disconnecting the transformer input and amplifying it through the capacitive-sensing backend are shown in Figure 10. The noise of transformer 1 measures approximately 19.7 uV/rtHz, while the noise of transformer 2 exhibits an 11% reduction to approximately 17.5 uV/rtHz, which is closely aligned with the anticipated theoretical calculations.

The absence of a JFET in the TIA circuit results in a higher amplifier current noise, and thus the manifestation of a transformer noise reduction is not adequately demonstrated. After optimizing the TIA circuit, the noise of transformer 2 exhibited a decrease of 73% when compared to that of transformer 1. Therefore, when the inherent noise of the TIA circuit is mitigated, the contribution from the transformer noise becomes pivotal in determining the overall circuit noise.

### 4.2. Experiment on Comparing Discrete TIA Noise

Based on the analysis presented in Section 3.1, the discrete TIA exhibits a significant potential for mitigating the effect of the amplifier current noise. In the manufacturing process of TIA circuits, meticulous attention is given to PCB layout and routing, which results in the realization of an optimized discrete TIA circuit via numerous iterations.

The utilization of a JFET as an input stage in TIA circuits leads to a significant decrease in TIA noise, from 230.3 nV/rtHz to 34.71 nV/rtHz, which represents only 15% of the former value. The noise performance was verified in the configuration with the open TIA input, and the results are shown in Figure 11. After normalization to account for circuit gain discrepancies, the discrete TIA noise decreases from 41.1 uV /rtHz to 7.17 uV /rtHz, as listed in Table 4, which represents approximately 17.4% of its initial value and is closely aligned with theoretical calculations. The gain calibration curves are shown in Figure 12.

### 4.3. Capacitive Sensing Resolution Test

The resolution of the capacitive sensing is directly proportional to the output noise of the capacitive sensing circuit, where the gain of the circuit serves as the proportionality factor. The carrier amplitude is set to 1.2 Vp and the input of the capacitive sensing circuit is provided by the TM simulator. As shown in Figure 13a, an air capacitor consisting of two copper sheets served as the input. Following calibration using an AH2700A capacitor bridge, the resulting tolerance was 4.6 fF. The test involved the utilization of transformer 2 and a discrete TIA, and the corresponding experimental setup is shown in Figure 13b.

As shown in Figure 14, the capacitive sensing resolution test result displays a value of 0.84 aF/rtHz@10 mHz, thereby exhibiting an enhancement of approximately 23% compared to the unoptimized measurement of 1.095 aF/rtHz@10 mHz [19].

The ground test results and in-flight test results from the LISA Pathfinder were 0.64 aF aF/rtHz @10 mHz and 0.7–1.8 aF/rtHz @10 mHz, respectively [16,18]. There is still a gap between our results and the LISA Pathfinder ground test results, which is mainly due to the influence of the excitation-signal amplitude stability. This paper mainly focuses on the optimization and testing of front-end circuit noise, and the optimization and analysis of the excitation-signal amplitude stability will be reflected in subsequent articles from our team.

## 5. Conclusions

This study investigated and optimized the preamplification circuit noise of a capacitive sensing circuit, thereby enhancing the resolution of the capacitive sensing in the intermediate and low-frequency ranges for space gravitational wave detection. In this study, we analyzed the effect of transformer performance on the pre-amplifier circuit and compared the equivalent noise of two transformers with different parameters. The results demonstrate that an increase in the inductance and quality factor decreases the equivalent noise of the transformer. The noise model of the TIA circuit was analyzed, and a discrete TIA was employed to mitigate the operational amplifier current noise as a noise source in the TIA circuit while supplementing the calculation method for drain-source channel resistance. This study examined the impact of the feedback capacitance and resistance on the noise in TIA circuits. These findings demonstrate that increases in feedback capacitance and resistance can effectively reduce the circuit background noise. However, it should be noted that augmenting the feedback capacitance may lead to a decrease in TIA gain, thereby necessitating comprehensive consideration in practical applications. Increasing the feedback resistance can reduce the noise in the discrete TIA circuit, but in conjunction with the inherent noise of the transformer, the reduced TIA noise cannot be well reflected, and the noise contribution of the transformer in the capacitive sensing circuit is approximately 80%. The performance of the transformer emerges as a pivotal factor that affects the noise characteristics of the preamplifier circuit in a capacitive sensing system.

The experimental results demonstrate that in the absence of JFET implementation as the TIA input stage, the equivalent noise level of transformer 2 is approximately 90% of that of transformer 1, which is consistent with the theoretical calculations. The optimized discrete TIA noise level was only 15% of its initial value, which aligned with the theoretical analysis. Finally, transformer 2 was employed to evaluate the resolution of the capacitive sensing in the presence of a discrete TIA. In comparison with previous experimental findings, the resolution of the capacitive sensing increased by approximately 23%, from 1.095 aF/rtHz at 10 mHz to 0.84 aF/rtHz at 10 mHz. This enhancement contributes to the low-frequency detection capability of space gravitational wave detection in space.

## Figures and Tables

**Figure 1 sensors-24-00805-f001:**
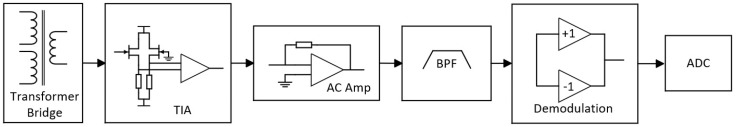
Schematic diagram of capacitive sensing circuit.

**Figure 2 sensors-24-00805-f002:**
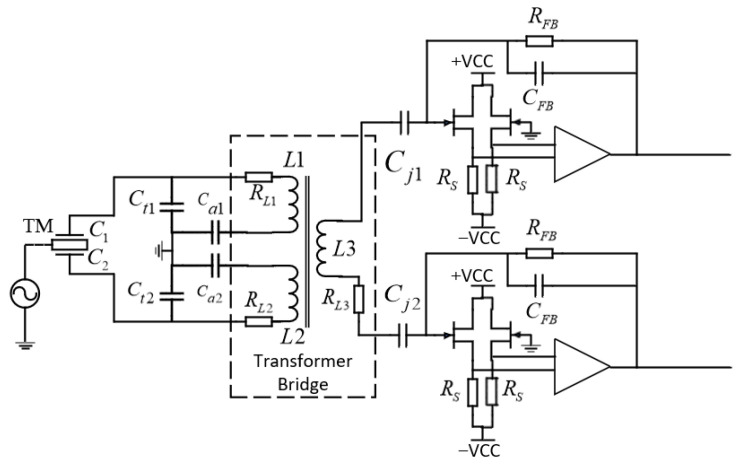
Capacitive sensing circuit front-end amplifier circuit.

**Figure 3 sensors-24-00805-f003:**
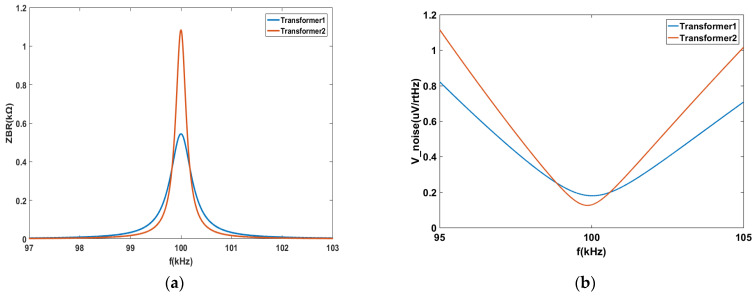
(**a**) The bridge’s equivalent impedance for the two transformers; (**b**) Equivalent noise of transformer 1 and transformer 2.

**Figure 4 sensors-24-00805-f004:**
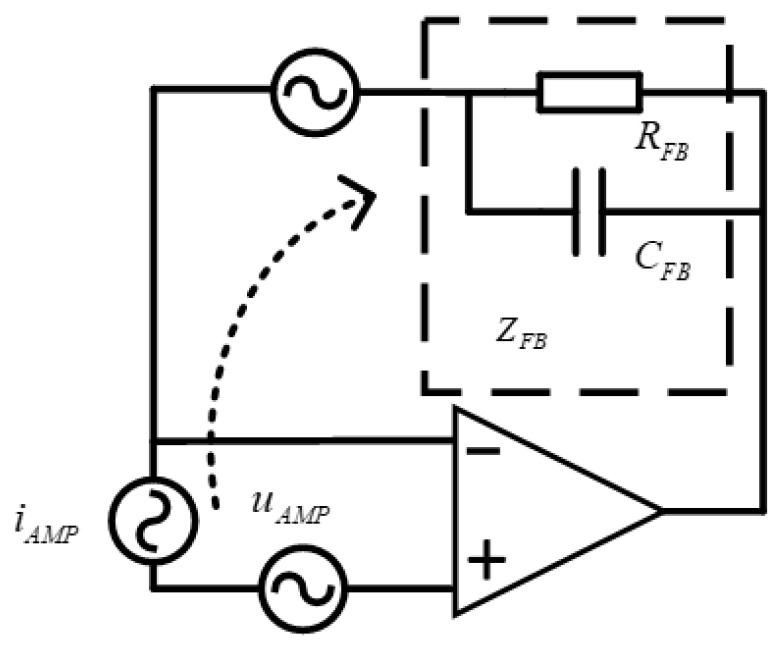
Decomposition of TIA circuit noise.

**Figure 5 sensors-24-00805-f005:**
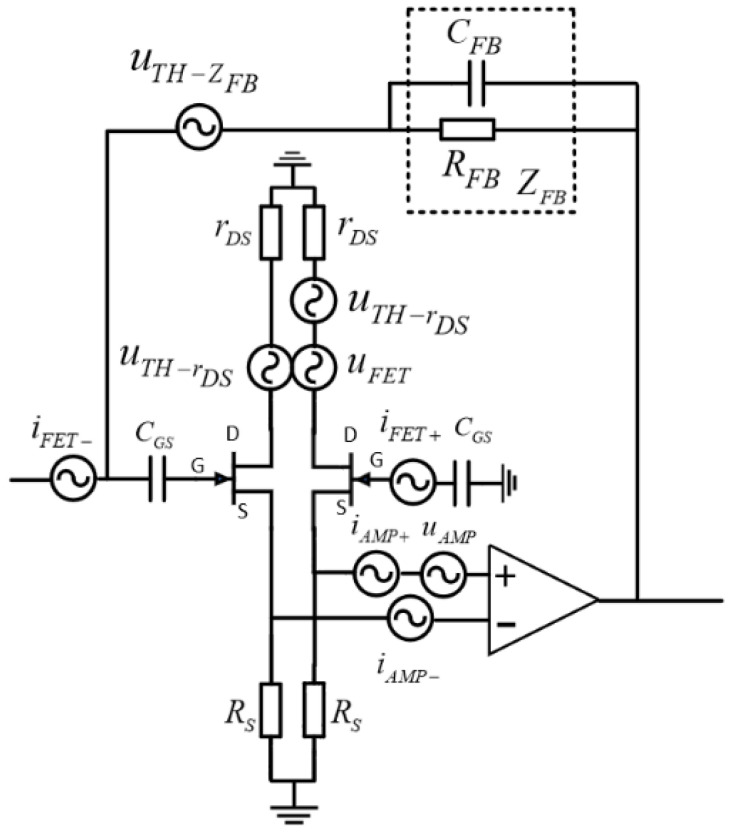
Decomposition of the TIA circuit noise utilizing a JFET as an input stage.

**Figure 6 sensors-24-00805-f006:**
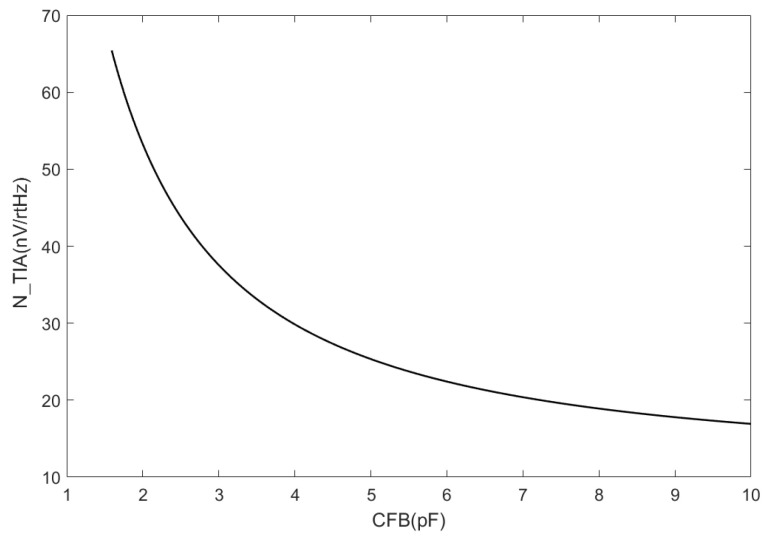
TIA noise decreases as the feedback capacitance increases.

**Figure 7 sensors-24-00805-f007:**
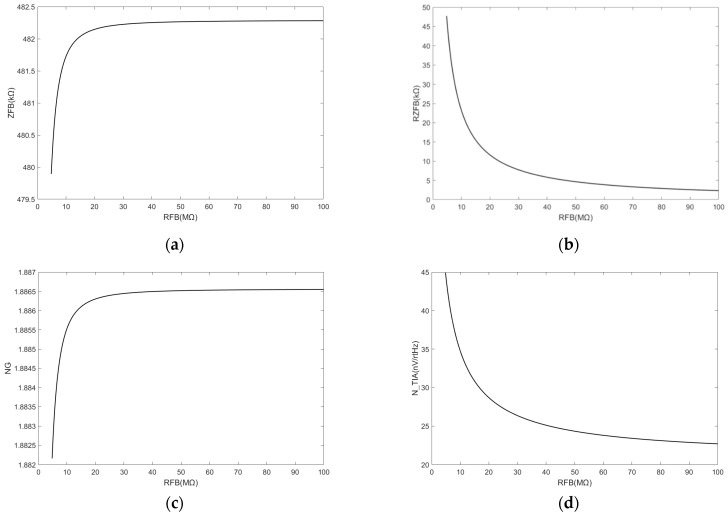
(**a**) Increase in feedback resistance leads to a corresponding increase in the feedback impedance, although the magnitude of the change is minimal; (**b**) The real part of the feedback impedance decreases when the feedback resistance increases; (**c**) Increase in the feedback resistance leads to a corresponding increase in noise gain, although the magnitude of the change is minimal; (**d**) The TIA noise decreases when the feedback resistance increases based on the real part of the feedback impedance.

**Figure 8 sensors-24-00805-f008:**
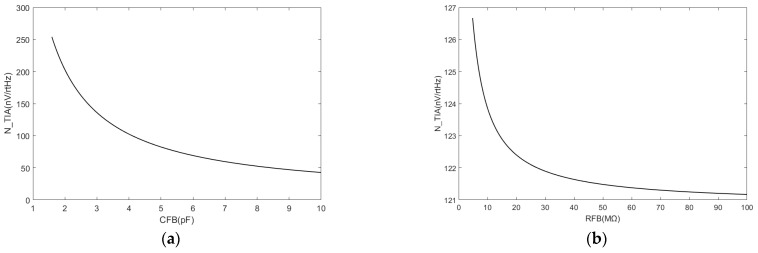
(**a**) TIA noise decreases when the feedback capacitance increases; (**b**) TIA noise decreases when the feedback resistance increases, although the variation remains within a range of less than 6 nV/rtHz.

**Figure 9 sensors-24-00805-f009:**
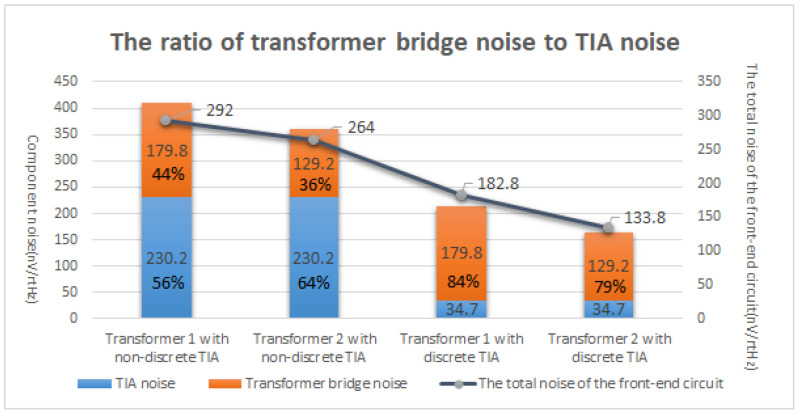
The ratio of transformer bridge noise to TIA noise.

**Figure 10 sensors-24-00805-f010:**
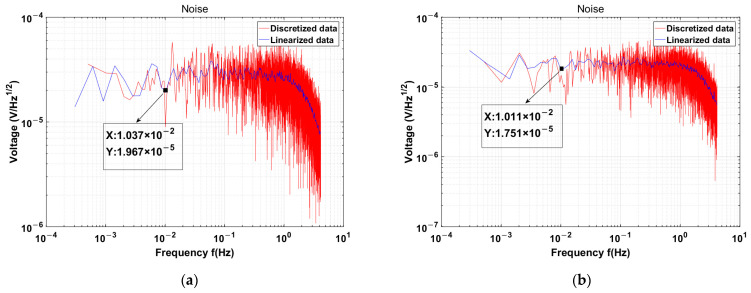
(**a**) Measurement result of transformer 1 is 19.7 uV/rtHz@10 mHz; (**b**) Measurement result of transformer 2 is 17.5 uV/rtHz@10 mHz.

**Figure 11 sensors-24-00805-f011:**
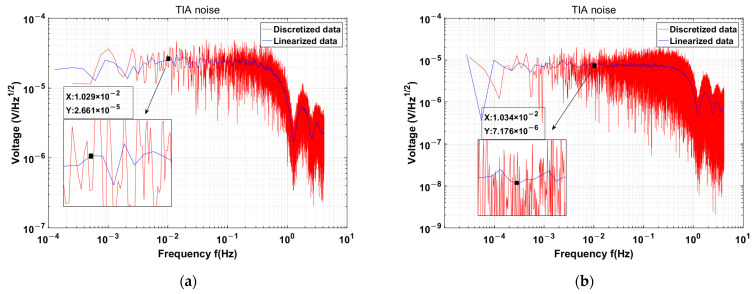
(**a**) TIA noise is 26.7 uV/rtHz with 26 V/pF, (**b**) Discrete TIA noise is 7.17 uV/rtHz with 40 V/pF.

**Figure 12 sensors-24-00805-f012:**
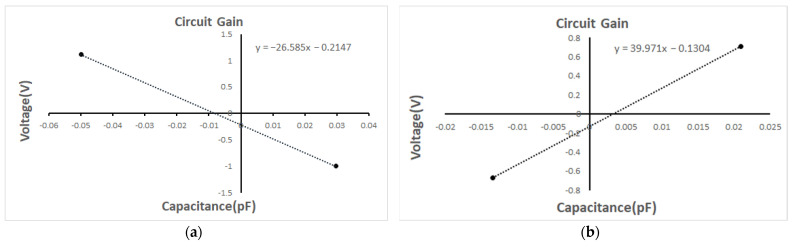
(**a**) The non-discrete TIA circuit gain test result is 26 V/pF, (**b**) The discrete TIA circuit gain test result is 40 V/pF.

**Figure 13 sensors-24-00805-f013:**
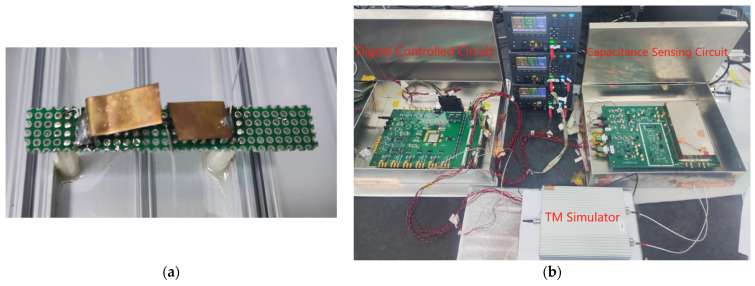
(**a**) TM simulator; (**b**) Test environment.

**Figure 14 sensors-24-00805-f014:**
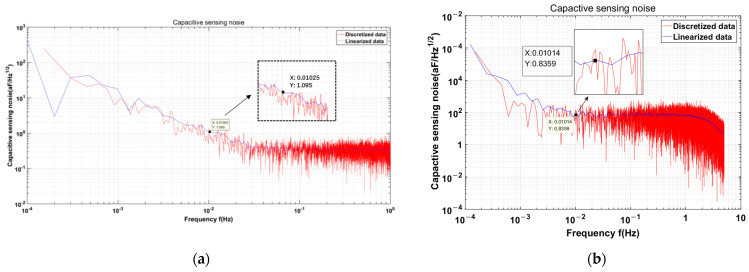
(**a**) The capacitive sensing resolution test using transformer 2 and a non-discrete TIA is 1.095 aF/rtHz at 10 mHz [19]; (**b**) The capacitive sensing resolution test result is 0.84 aF/rtHz at 10 mHz using a discrete TIA with transformer 2.

**Table 1 sensors-24-00805-t001:** Pre-amplifier circuit parameters.

Parameter	Transformer 1	Transformer 2
Inductor (L)	4.4 mH	4.37 mH
Quality factor (Q)	197	395
Coupling factor (K)	0.97	0.97
Tuning capacitors ($C_{t1}$,$C_{t2}$)	304.68 pF	306.85 pF
Driving capacitors ($C_{a1}$,$C_{a2}$)	10 nF	10 nF
$C_{FB}$	3.3 pF	
$k_{B}$	1.38 × 10^−23^ m^2^kgK^−1^ s^−2^	
T	300 K	
f	100 kHz	

**Table 2 sensors-24-00805-t002:** Characterization of circuit parameters and analysis of noise calculations.

Parameter	Value	Noise Source	Equation	Result
$u_{AMP}$	5 nV/rtHz	AMP voltage noise	$\sqrt{2}\times u_{AMP}NG$	13.3 nV/rtHz
$i_{AMP}$	335 fA/rtHz	AMP current noise	$\sqrt{2}\times i_{AMP}\left|Z_{FB}\right|$	228.2 nV/rtHz
$C_{FB}$	3.3 pF	TIA thermal noise	$\sqrt{2}\times \sqrt{4k_{B}Tℜ\left[Z_{FB}\right]}$	27.73 nV/rtHz
$R_{FB}$	10 MΩ	Total noise		230.3 nV/rtHz

**Table 3 sensors-24-00805-t003:** Results of noise calculations for the discrete TIA.

Noise Source	Equation	Result
TIA voltage noise	$\sqrt{2}\times u_{AMP}NG$	13.3 nV/rtHz
TIA current noise	$2\times i_{AMP}r_{DS}NG$	0.13 nV/rtHz
TIA thermal noise	$\sqrt{2}\times \sqrt{4k_{B}Tℜ\left[Z_{FB}\right]}$	27.7 nV/rtHz
JFET voltage noise	$2\times u_{FET}NG$	15.1 nV/rtHz
JFET current noise	$\sqrt{2}\times i_{FET-}\left|Z_{FB}\right|$	2.9 nV/rtHz
JFET current noise	$\sqrt{2}\times i_{FET+}r_{DS}NG$	0.001 nV/rtHz
JFET thermal noise	$2\times \sqrt{4k_{B}Tr_{DS}}NG$	4.9 nV/rtHz
Total noise		34.71 nV/rtHz

**Table 4 sensors-24-00805-t004:** Results of TIA noise normalization.

	Noise	Gain	Normalized Result
TIA	26.7 uV/rtHz	26 V/pF	41.1 uV/rtHz
Discrete TIA	7.17 uV/rtHz	40 V/pF	7.17 uV/rtHz

## Data Availability

The datasets presented in this article are not readily available because the data are part of an ongoing study or due to technical limitations. Requests to access the datasets should be directed to chenyz_mail@163.com.

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
