# Peer review of "Research and Optimization of High-Performance Front-End Circuit Noise for Inertial Sensors"

_sensors, 2024, doi:10.3390/s24030805_

Round 1

Reviewer 1 Report

Comments and Suggestions for Authors

The work lacks originality, which is similar to the paper named "Development of Electronic System for Sensing and Actuation of Test Mass of the Inertial Sensor LISA". The authors need to clarify the innovative and the contribution.

 1. The authors have already pointed out that the use of jfet to reduce current noise in front-end circuits is a more mature scheme, and the authors should describe what is innovative about the work in this paper? What are the advantages of the results of this paper compared with the peers? Why didn’t you use the low current noise op-amps such as LTC6268?

2. Is the noise modeling of a discrete jfet front-end circuit correct? The input voltage noise and input current noise of the op-amp will be induced to the gate through the gate source capacitance of the jfet. In Table 4 about each noise calculation and noise factor, the voltage and current noise of the op-amp does not flow directly through the feedback loop, why is the noise gain also NG?

3.Why is the Driving Capacitance value in Table1 at the pF level?

4. What is the amplitude of the carrier wave? There is a lack of circuit related gain calibration tests.

5. What is the purpose of the section of 3.1.3?

6. The authors do not give experimental data curves for the tuning noise of the separated front-end circuit, similar to Figure3, to illustrate the percentage of the contribution of other noises such as transformer thermal noise, jfet, op-amps, etc. in the new scheme. Does the tuned output noise of the front-end circuit agree with the theoretical calculation?

Comments on the Quality of English Language

N.A.

Reviewer 2 Report

Comments and Suggestions for Authors

The paper describes a way to optimize the performances of a circuit to implement the capacitive sensing of a test mass motion in a high performances inertial sensor for the detection of gravitational waves in space.

The research topic is of clear interest but the paper requires major improvements to be considerable for publication. In particular:

- The text requires a careful revision. It is very often confused and the message is not properly conveyed in good English

- The introduction is missing to put the work in perspective of the decadal work performed by the LISA Pathfinder mission on the very same topic of the paper. References to papers describing the LISA Pathfinder inertial sensor electronics are missing. Ref 17, that describes a development very similar to the work reported in the paper, is mentioned only at page 4 without commenting on the content of that work.

- Pag 2. The description of the working principle of the system is confused and errors in the text are observed (e.g. TIA stands for trans-impedance amplifier)

- pag 2 row 82. A previous research is mentioned but the reference and a description of such research is missing

- pag 13 row 336. It is not clear how the 89% reduction is calculated. A 89% reduction with respect to 19.7 uV/sqrt(Hz) would make 2.17 uV/sqrt(Hz) and not 17.5 uV/sqrt(Hz)

- pag 13 row 347. It is not clear what is intended with the sentence "The TIA input terminal is suspended"

- pag 14 row 366. The measurement showing 1.095 aF/sqrt(Hz) is missing and it should be presented for comparison and completeness

Comments on the Quality of English Language

The text requires a careful revision. It is very often confused and the message is not properly conveyed in good English

Reviewer 3 Report

Comments and Suggestions for Authors

This manuscript reports a capacitive displacement sensing circuit research for TaiJi GW program. By introducing a discrete JFET into the circuit,the authors obtain a lower sensing noise compared with their previous result. The followings should be sufficiently addressed:

1. The authors should summarize the advantages and disadvantages of introducing a discrete JFET.

2. The authors should compare their results with other groups, such as LPF ground testing.

3. P1 L23  a reduction in capacitance sensing resolution?

4. P4 L136  The simulated and analyzed... , shouldn't there be four curves in Fig.3?

5. Some mathematic symbols used in this manuscript are inappropriate, such as the impedance and the real part

Round 2

Reviewer 1 Report

Comments and Suggestions for Authors

Thank you for your response.

Author Response

Thank you for your comments.

Reviewer 2 Report

Comments and Suggestions for Authors

The content of the paper improved since the previous review, but there are still items to be corrected:

Pag 2 line 53 - 54: "But there is still an order of magnitude gap between the minimum test frequency band and noise metrics required by the LISA Pathfinder design and the LISA mission[21]" That is an outdated information. As reported in Phys. Rev. Lett. 120, 061101 (2018) - Published 5 February 2018, LISA Pathfinder demonstrated performances at the level required for LISA in the LISA band

Pag 2 line 57 - 58: "However, the transformer noise has not been verified by actual measurement, and the influence of feedback resistance and capacitance on TIA noise has not been further analyzed." It is actually not true. Verification is presented in Ref 17.

Pag 14 line 362: "there is still a gap, mainly due to the influence of the amplitude stability of the excitation signal". This is a speculation of the authors not reflected in the content of the cited bibliography, therefore it shall be declared as an hypothesis.

Comments on the Quality of English Language

/
